# Direct Writing Corrugated PVC Gel Artificial Muscle via Multi-Material Printing Processes

**DOI:** 10.3390/polym13162734

**Published:** 2021-08-15

**Authors:** Bin Luo, Yiding Zhong, Hualing Chen, Zicai Zhu, Yanjie Wang

**Affiliations:** 1State Key Laboratory for Strength and Vibration of Mechanical Structures, Xi’an Jiaotong University, Xi’an 710049, China; 2School of Mechanical Engineering, Xi’an Jiaotong University, Xi’an 710049, China; ydzhong@zju.edu.cn (Y.Z.); zicaizhu@xjtu.edu.cn (Z.Z.); 3School of Mechanical and Energy Engineering, Shaoyang University, Shaoyang 422000, China; 4Changzhou Campus, School of Mechanical and Electrical Engineering, Hohai University, Changzhou 213022, China; yj.wang1985@gmail.com

**Keywords:** direct writing, PVC gel, artificial muscle, rheological behavior, integrated printing

## Abstract

Electroactive PVC gel is a new artificial muscle material with good performance that can mimic the movement of biological muscle in an electric field. However, traditional manufacturing methods, such as casting, prevent the broad application of this promising material because they cannot achieve the integration of the PVC gel electrode and core layer, and at the same time, it is difficult to create complex and diverse structures. In this study, a multi-material, integrated direct writing method is proposed to fabricate corrugated PVC gel artificial muscle. Inks with suitable rheological properties were developed for printing four functional layers, including core layers, electrode layers, sacrificial layers, and insulating layers, with different characteristics. The curing conditions of the printed CNT/SMP inks under different applied conditions were also discussed. The structural parameters were optimized to improve the actuating performance of the PVC gel artificial muscle. The corrugated PVC gel with a span of 1.6 mm had the best actuating performance. Finally, we printed three layers of corrugated PVC gel artificial muscle with good actuating performance. The proposed method can help to solve the inherent shortcomings of traditional manufacturing methods of PVC gel actuators. The printed structures have potential applications in many fields, such as soft robotics and flexible electronic devices.

## 1. Introduction

Artificial muscle is a material or structure that can undergo various forms of deformation under external excitations (i.e., electricity [1,2,3,4,5], heat [6,7,8], light [9,10], magnetic field [11,12], fluid pressure [13,14,15], and chemical stimulation [16]) and output strain and stress. Artificial muscle is named for its ability to achieve biological muscle-like function, and it has broad application prospects in fields such as robots [4,13,17,18,19], flexible electronics [20,21], smart textiles [22], and medical rehabilitation [23,24]. Among the various actuating modes of artificial muscles defined according to the external excitation, the electrical actuating mode has advantages of easy control and a wide application range. Electroactive polymers (EAPs) are a class of electrically actuated artificial muscle materials that are of great interest to researchers. Typical representatives of EAPs include ionic polymer/metal composites (IPMCs), dielectric elastomers (DEs), and electroactive polyvinyl chloride (PVC) gel. Among these materials, IPMC has advantages of low operating voltage (1–3 V) and a response time of seconds; however, its actuating force is limited, the working energy density is less than other EAPs, and its durability is poor in dry environments [25]. DE has the advantage of a fast response and can work in air; however, DEs generally require operating voltages of up to several kV to meet electric field strength requirements, which increases the risk of electrical breakdown. However, PVC gel has advantages of high output stress and strain, quick response, good thermal stability, moderate working voltage, low power consumption, and a long cycle life [5], as well as the advantages of DE and IPMC. Therefore, it has more potential in the application [26].

Deformation principle of PVC-gel actuator is shown in Figure 1; in an applied electric field, PVC molecules migrate to the anode, the gel is polarized, and a Maxwell force is generated, which results in creep deformation of PVC gel near the anode and compression of PVC gel in the thickness direction. The PVC gel can return to its original shape under its own elasticity when the electric field is turned off [27].

The PVC-gel actuators in application include the traditional mesh anode-planar PVC gel-planar cathode structure (referred to as the planar PVC gel actuator) [5,24] and, more recently, the planar electrode-corrugated PVC gel structure (referred to as the corrugated PVC gel actuator) [20,28]. For any of the structures, the basic actuating unit is the sandwich structure as electrode layer–core layer–electrode layer. A key common feature of the two typical actuators is the existence of pore structures, which are necessary for the actuators to deform effectively; for example, the migration of the peak material to the valley floor or the transfer of planar materials into the mesh pores.

The current manufacturing method of PVC gel is to mix PVC, dibutyl adipate (DBA), and tetrahydrofuran (THF) together to form a precursor material and then cast the material into molds to obtain films with designed shapes [5]. To make a PVC gel actuator, the traditional method also requires the fabricated PVC gel films to be cut and then manually stacked with metal electrodes. Obviously, the casting method is only suitable for manufacturing simple 2D structures, and it is difficult to produce complex, diversified, flexible 3D structures. To overcome this issue, additive manufacturing is a possible choice. Rossiter and co-workers proposed a filament additive manufacturing technique to create complex (3D) PVC gel structures. The 3D printing process was introduced, where a precursor material (PVC precursor powder mix with DIDA) is extruded into a thermoplastic filament for 3D printing [29]. However, the electrode printing process results in many difficulties because most electrode materials are not thermoplastic, which is essential for PVC-gel as actuators.

Recently, multiple reports have shown that a wide variety of materials have been used in 3D direct writing, including hydro-gels [30], nano-particles [31], polyelectrolytes, ceramics [32], shape memory polymers, and carbon materials [33]. Direct writing is a layer-by-layer assembly technique in which shear-thinning inks are extruded through a nozzle in a programmable pattern, upon which the inks rapidly solidify via gelation, evaporation, or temperature-induced phase change [32]. There are few reports with respect to PVC-gel solution printing by direct writing. Meanwhile, electrode printing, when adopting the direct writing process, has been successful in the presence of a conductive solution that can be cured, such as carbon nanotubes (CNTs) doped polymer and ionic gels [33]. In this work, integrated printing process of electrodes and core layers for fabricating PVC gel artificial muscle were proposed. CNTs doped shape memory polymers (CNT/SMP) were designed for PVC-gel electrode. Sacrificial materials (F127) were used for creating different PVC core structure. Inks with suitable rheological properties were developed for printing four functional layers, including core PVC layers, electrode layers, sacrificial layers, and insulating layers, with different characteristics. The corrugated PVC gel structural parameters were designed and optimized to improve the actuating performance of the PVC gel artificial muscle. Several techniques were used to evaluate the feasibility of the process, including conductivity tests, rheology measurements, and electro-mechanical tests.

## 2. Experimental Section

### 2.1. Materials Preparation and Structure Design

Polyvinyl chloride (PVC, degree of polymerization 4000) was purchased from Scientific Polymer Products Inc., New York, NY, USA. Dibutyl adipate (DBA) was obtained from Hubei Jusheng Technology Co., Ltd., Tianmen, P.R. China. Tetrahydrofuran (THF) was purchased from Tianjin Fuyu Fine Chemical Co., Ltd., Tianjin, P.R. China. Pluronic F-127 was purchased from Sigma-Aldrich, St. Louis, MO, USA. The multi-walled CNTs (Time Nano China, TSW3, diameter 10–20 nm, length 0.5–2 μm, >98%) were from Chengdu Organic Chemicals Co., Ltd., Chinese Academy of Sciences, Beijing, P.R. China. Dimethylacetamide (DMAC) was obtained from Aladdin Reagent Co., Ltd., Shanghai, P.R. China. Shape memory polymer (SMP, DiAPLEX MM4520) was from Mitsubishi Co., Ltd., Tokyo, Japan. Ecoflex 00-20 was purchased from Smooth-On Inc., Macungie, PA, USA.

As shown in Figure 2a, CNT doped shape memory polymers (CNT/SMP) were chosen for printing PVC-gel electrode because the electrodes’ hardness is suitable [31]. The Ecoflex silicone materials, which possess very high insulation performance, were designed for connecting PVC-gel actuator units. To fabricate the flexible artificial muscle structure, it was first necessary to prepare inks suitable for direct writing and to satisfy the functional requirements for core layers, sacrificial layers, electrode layers, and insulating layers. Using a printing device with a pneumatic ink extrusion system (Appendix A), each functional layer was printed in sequence with the prepared inks to produce the composite structure. In particular, pluronic F-127 is a polyethylene oxide (PEO)-polypropylene oxide (PPO)-polyethylene oxide (PEO) triblock copolymer. As shown in Figure 2b, F-127 has thixotropic properties. When the ink was prepared, the molecules cross-linked with each other to form gel networks. When the gel was subjected to shear stress, the networks broke down and the ink became fluid-like so it could be conveniently extruded through a nozzle. After the shear stress was removed, the ink quickly turned to gel with rigidity that could help maintain the shape of the printed structure. In this paper, the F-127 gel inks were printed for constructing a corrugated PVC-gel core structure.

The PVC gel ink consisted of three constituent materials: PVC, DBA, and THF. For preparation of the gel ink, the three constituents were proportionally mixed (referring to research on the relationship between the proportion of PVC gel and its properties in literature [34,35], the mass ratio used in this paper was PVC:DBA:THF = 1:7:12), and then the solution was stirred by a magnetic stirrer for 2–3 days until the gel became homogenous, colorless, and transparent. F-127 gel ink was comprised of F-127 and deionized water. During the preparation process, F-127 was mixed with deionized water in proportion and then the mixture was stirred for 15 min at 2000 rpm by a planetary mixer (HM800) to obtain the colorless, transparent gel. When preparing CNT/SMP composite ink, SMP was first mixed with DMAC, which served as a solvent, and the mixture was heated (60 °C) and magnetically stirred (600 rpm) for approximately 4 h such that the SMP particles were completely dissolved to form a transparent solution. Subsequently, CNTs were added proportionally to the solution and the mixture was dispersed by ultrasound for approximately 3 h until the carbon nanotubes were evenly dispersed in the solution. Finally, using a magnetic stir heater (60 °C, 600 rpm), the solution was concentrated for approximately 5 h to produce ink containing 30 wt.% solid phase that could be used for printing. Ecoflex silicone consisted of two components: Ecoflex-A and Ecoflex-B. To prepare the Ecoflex ink, the A and B components were mixed at a mass ratio of 1:1 and stirred.

### 2.2. Performance Characterization of the Inks

When analyzing the electrical properties of the CNT/SMP composites, CNT/SMP composites with different CNT contents were made into long strips (40 mm × 5 mm × 1 mm). The resistance of the strips was measured using a source meter (2450 SourceMeter, Keithley, Tektronix, Cleveland, OH, USA) and the conductivity of composites with different proportions was calculated. The rheological behaviors of the prepared inks were measured by a rheometer (MCR 302, Anton Paar, Graz, Austria) fitted with a parallel-plate geometry (PP35Ti, diameter of 35 mm and gap of 0.6 mm) at 25 °C. The apparent viscosity, storage modulus (G′), and loss modulus (G″) of the PVC gel ink, F-127 gel ink, CNT/SMP composite ink, and Ecoflex ink were measured as a function of the angular frequency from 0.628 rad/s to 62.8 rad/s under fixed strain (1%) and shear stress (from 0.1 to 100 Pa) in oscillatory mode at 1 Hz.

### 2.3. Direct Writing Process

In this paper, the 3D printing used a multi-nozzle direct writing device (Appendix A). In the first step, the 3D models were established in CAD software (SolidWorks 2016) and then imported software to generate G codes. The second step was to install the syringe with ink for printing to the designed position in the direct writing device. In the third step, the designed structure was printed layer-by-layer by controlling the movement of the 3D positioning stage according to G codes and adjusting the output air pressure to control the extrusion speed of the ink.

### 2.4. Post-Processing of the Printed Structures

After printing the composite structures of the PVC gel actuators, it was necessary to immerse them in water to remove the sacrificial material and obtain the designed corrugated PVC gel. Post-processing by curing was also needed for the printed CNT/SMP composites (for both the curing of the shape and the formation of conductive network characterized by improved conductivity). Therefore, the curing characteristic of the CNT/SMP composite ink and the relationships between the conductivity and time at different curing temperatures were studied. We used 10 wt.% CNT/SMP ink to produce rectangular sheet samples (40 mm × 20 mm × 0.4 mm) that were heated in a vacuum oven or stored at room temperature. At regular intervals, the shape curing was observed, and the resistance was measured to determine the relationship between the conductivity and time. To study integrated printing of PVC gel artificial muscles, we combined the above two post-processing methods, proposing a method for placing the printed structure at room temperature first and then heating in a water bath, and studied the effects of this method on the dissolution of the sacrificial layers and the curing of the CNT/SMP composite ink.

### 2.5. Performance Characterization of the PVC Gel Artificial Muscle

The actuating performance of the PVC gel artificial muscle was measured by a test system (Appendix A). The displacement of the artificial muscle with a laser sensor (LK-G500, KEYENCE, Osaka, Japan) was measured using a signal generator (DG4062, RIGOL, Beijing, P.R. China) and voltage amplifier (MODEL 20/20C, TREK, Waterloo, WI, USA) to apply voltage to the electrode layers of the artificial muscle.

## 3. Results and Discussion

### 3.1. Printability of Printing Inks

Rheology measurements were conducted for evaluating printability of inks. As shown in Figure 3a,b. The apparent viscosity of the prepared inks decreased with an increase in the angular frequency, proving that they were shear-thinning non-Newtonian fluids appropriate for direct writing. The G′ of the PVC gel ink was greater than its G″, and their difference decreased with an increase in angular frequency.

Therefore, when the shear effect was strong and the angular frequency was high, the PVC gel ink had excellent fluidity and, thus, good filling performance. Alternatively, when the shear effect was weaker and the angular frequency decreased, the difference between G′ and G″ increased, which resulted in a decrease in fluidity; therefore, the printed shape was easy to maintain. These characteristics are also demonstrated in Figure 4a. The PVC-gel ink exhibited solid-like gel behavior (G′ > G″) at stresses (0.1–29.81 Pa) and liquid-like gel behavior (G′ < G″) at stresses >29.81 Pa, which indicated that this ink is suitable for direct writing [36]. As for F-127 gel ink, at the same angular frequency, with an increase in F-127, the viscosity of the F-127 gel ink increased, and the printability increased. The G′ of each experimental group was greater than G″, indicating that the material was in a non-flowing state. With the increase in F-127 content, this behavior was more obvious, and thus, the shape retention ability of the printed structure and the printing accuracy were better. Additionally, 40 wt.% F-127 gel ink storage and loss modulus decreased steadily when the stress was increased from a very low stress of 0.1 Pa up to 10 Pa in Figure 4b, and the 40 wt.% F-127 gel ink exhibited solid-like gel behavior (G′ > G″) at stresses (0.1–7.41 Pa) and liquid-like gel behavior (G′ < G″) at stresses >7.41 Pa, which indicated the thixotropic properties. As the main purpose of using this material was to obtain the required pore structure in the core layers with high printing precision, in the following study, 40 wt.% F-127 gel ink was used to print the sacrificial layers.

The rheological properties of the Ecoflex ink were measured after storage at room temperature (25 °C) for 20 min. The ink reached a semi-cured state since the molecular chains of the two components had gradually crosslinked, and the G′ of the Ecoflex ink was greater than G″. Therefore, the ink can be used to print stable lines. For both pure SMP ink and CNT/SMP ink with 10 wt% carbon nanotubes, their G″ were greater than their respective G′, demonstrating strong fluidity suitable for printing the planar structure in this paper that can also produce good surface quality. Additionally, after CNT loading, the viscosity and moduli (G′ and G″) of the ink were greater than that of pure SMP ink, showing that CNT had a good thickening effect. The 10 wt% CNT/SMP composite ink’s fluidity is indicated in Figure 4d; its loss modulus was higher than the storage modulus at stresses (0.1–10 Pa). We systematically evaluated the printability of inks by process experiments. The influence of the extrusion pressure (P) and nozzle moving speed (V) on the filament width of the printed inks are listed in Figure 5. At the lowest extrusion pressure and highest nozzle moving speed, a minimum fiber diameter was achieved with the nozzle diameter (0.29 mm) for each ink. The process parameters used in the subsequent printing experiment are as follows: PVC-gel inks (P = 20 KPa, V = 12 mm/s); F127-gel inks (P = 370 KPa, V = 12 mm/s); CNT/SMP inks (P = 260 KPa, V = 6 mm/s); Ecoflex inks (P = 180 KPa, V = 10 mm/s).

### 3.2. Research Results of Curing Methods

The solidification of electrode and the integrated structure shown in Figure 2 is the key to the success of printing PVC-gel actuators. The curing properties of the CNT/SMP ink were analyzed (including both shape curing and conductive network formation characterized by improved conductivity) at different temperatures. After heating at 80 °C for 10 min, 60 °C for 15 min, or storage at room temperature (25 °C) for 48 h, the shape of the composites was completely cured and their surface was uniform and dense. The conductivity–time relationships of the CNT/SMP composite ink at different temperatures are shown in Figure 6a,b. As shown in the figure, higher temperatures resulted in a faster increase in conductivity and a higher final conductivity. After heating at 80 °C for 3 h, the conductivity tended to be stable up to 5.06 S/m. After heating at 60 °C for 4 h, the conductivity reached 3.61 S/m. At room temperature, the curing speed was too slow to meet the requirements of this paper. However, because of compatibility issues of multiple different material systems, the layer-by-layer thermal curing method was not suitable for the integrated printing of multi-material composite structures. For example, the thermal curing of the upper electrode layer may cause the sacrificial layer to lose moisture quickly and melt before the shape of the upper electrode layer is solidified, leading to collapse of the overall structure. Similarly, repeated heating at high temperature for a long time may also affect the performance of PVC gel core layers. Therefore, we proposed a step-wise curing method for the printed electrode layer, which included room temperature storage for 48 h to cure the shape and then uniform heat in a water bath after printing. Uniform heating was used to avoid the effects of repeated heating on the performance of the PVC gel core layers. Water bath heating allowed the sacrificial material to be rapidly dissolved and helped uniformly heat the electrode. Regarding the heating temperature, long-term heating of the structure at 80 °C could greatly attenuate the performance of PVC gel core layers, while heating the structure at 60 °C had little effect. Therefore, 60 °C was selected as the heating temperature. For the printed electrode layers, the first step of incubating at room temperature for 60 h was to ensure that the shapes of the samples were completely cured. As shown in Figure 6c, the conductivity increased slowly at room temperature, and increased rapidly when the samples were heated in a water bath. After the samples were heated in a water bath at 60 °C for 30 min, the conductivity reached 1.87 S/m, which meets the requirements for conductivity of the electrode material in actuators. Therefore, when separately printing the electrode layers, the selected curing condition was 80 °C for 3 h. For integrated printing of PVC gel actuators, the curing method for the electrode layers was first storage at room temperature for 60 h until the shape of the newly printed electrode layer was completely cured, printing of the other layers, and after the shape of the top electrode layer was completely cured, heating of the entire structure in a 60 °C water bath for 30 min.

### 3.3. 3D Printing Processes of Flexible PVC Gel Artificial Muscle Structures

The printing methods of the core layers, electrode layers, and whole artificial muscles were studied. Initially, the printing method of the corrugated PVC gel core layers was studied. A corrugated F-127 gel sacrificial layer was printed and then the PVC gel was printed on top, filling the pores of the sacrificial layer. The PVC gel was cured at room temperature (25 °C) for approximately 12 h. The resulting composite structure was immersed in water to remove the sacrificial material, and after drying, a corrugated PVC gel core layer was produced, as shown in Figure 7a. The fabricated corrugated PVC gel core layer had good elasticity and transparency with an obvious corrugated shape. The shape of the corrugation in the core layers is shown in Figure 7b. To study the effect of the corrugation scale on the actuating performance of the PVC gel core layers, we designed and printed five groups of corrugated PVC gels with different spans, b. The dimensions of each group were as follows: a = 0.2 mm, h = 2 mm, d = 0.3 mm, and b = 0.8, 1.2, 1.6, 2.0, 2.4 mm. The bottom of the core layer had a planar size of 40 mm × 20 mm.

Additionally, we printed CNT/SMP composite electrode layers (40 mm × 20 mm × 0.2 mm) and then cured them at 80 °C for 3 h. Stainless steel wire and conductive silver paste were used to wire them together. As shown in Figure 7c, the finished electrode layer had a dense and smooth surface with suitable rigidity.

Based on the above research, we printed the artificial muscle structure shown in Figure 2a in an integrated way. From the perspective of manufacturing, the functional layers of this structure were integrated printings layer-by-layer at different times to produce a complete structure. Regarding the working principle, all of the electrode layers in contact with the corrugated side of the corrugated PVC gel core layers should be connected to the anode, while all of the electrode layers in contact with the planar side of the corrugated PVC gel core layers should be connected to the cathode. To avoid mutual interference or coupling of different actuating units when powered, the insulating layers of silicone Ecoflex were placed between adjacent actuating units.

As shown in Figure 7d, single layer actuators (actuating units) were first printed. At the beginning, the bottom electrode layer was printed and then heated in a vacuum oven at 80 °C for 3 h to completely cure. Subsequently, the sacrificial layer was printed over the bottom electrode layer. The PVC gel core layer was then printed on the sacrificial layer to fully fill and cover the pores of the sacrificial layer, and the sample was stored at room temperature for approximately 12 h until the THF was sufficiently volatilized and the PVC gel was cured. The top electrode layer was then printed on the core layer, and the sample was stored at room temperature for 60 h until the top electrode layer was cured. After printing, the following post-processing of the sample was performed. First, the composite structure was heated in a water bath at 60 °C for 30 min until the sacrificial layer was completely dissolved. Next, after drying, the edges of the sample were cut to avoid contact between the two electrode layers at the edges, which may result in short circuit of the actuator during testing. Finally, stainless steel wire and conductive silver paste were used to create a lead from the two electrode layers. After heating at 60 °C for approximately 15 min to solidify the conductive silver paste, the integrated printed corrugated PVC gel actuating unit was complete.

On this basis, we studied the printing method of a multilayer PVC gel actuator (two layers and three layers), and Figure 7e illustrates the printing process with a triple layer actuator as an example. Insulating layer 1 was printed on actuating unit 1 and then stored at room temperature for 4 h for complete cure. Similarly, the bottom electrode layer, sacrificial layer, core layer, and top electrode layer were printed in sequence, wherein the electrode layers and core layer were both cured at room temperature before printing the next functional layer, thereby, completing the printing of actuating unit 2. The process continued to superimpose the required number of actuating units. After the overall structure was printed and the electrode layers cured, the entire structure was heated in a water bath at 60 °C for 30 min to remove the sacrificial layer. The sample was then trimmed and leaded to complete fabrication of the multilayer actuators.

### 3.4. Performance Characterization of the Printed Structures

To test the actuating performance of the printed corrugated PVC gel core layers, three pieces of PVC gel from each of the five experimental groups with varying spans and four pieces of zinc foil electrode (40 mm × 20 mm × 0.07 mm) were superimposed to make a metal electrode-based PVC gel actuator. At different voltages (400 to 800 V, 1 Hz, 50% duty cycle square wave), the strain of the actuators was tested as a function of the corrugated span of core layers, as shown in Figure 8a. The strain of each actuator increased with an increase in voltage. As the corrugated span increased, the strain at each voltage exhibited a single peak form that increased first and then decreased. The corrugated PVC gel with a span of 1.6 mm had the best actuating performance, and the strain of the actuator using this kind of PVC gel core layer reached 9.9% at 800 V. Overall, the PVC gel printed using the additive manufacturing method described in this paper had good electro–deformation performance. Additional experiments in this study used a corrugated PVC gel with a span size 1.6 mm.

To prove the applicability of the printed CNT/SMP composite electrode, we superimposed three pieces of corrugated PVC gel with the 1.6 mm span and four pieces of CNT/SMP composite electrode to form a CNT/SMP composite electrode-based PVC gel actuator and tested the strain versus voltage (400 to 800 V, 1 Hz, 50% duty cycle square wave). Figure 8b shows a comparison of the actuating performance of the CNT/SMP composite electrode-based actuator and a metal electrode-based actuator with the same kind of core layer. The strain of the two actuators at the same voltage were similar. The strain of the CNT/SMP composite electrode-based actuator reached 10.3% at 800 V, proving that the performance of the printed CNT/SMP composite electrode was similar to that of the metal electrode, meeting the requirements of this study.

For testing the actuating performance of the corrugated PVC gel actuators obtained by integrated printing, the strain–voltage (400 to 800 V, 1 Hz, 50% duty cycle square wave) relationship without load and the strain–load (0 to 70 g) relationship (constant application of 800 V, 1 Hz, 50% duty cycle square wave) of the actuators with varied number of layers (1 to 3) were measured. As shown in Figure 8c, at the same voltage and as the number of layers increased, the strain gradually decreased: at 800 V, the strain of the single layer actuator was as high as 10.9%, the strain of the double layer actuator was up to 8.5%, and the strain of the triple layer actuator reached 8.0%. Since the insulating layers in the multilayer structure increased the total thickness, the strain of the actuators typically decreased to some extent as the number of layers increased. As the load increased, the strain gradually decreased, and the speed gradually slowed. The fewer number of layers, the faster the decreasing speed. When subjected to the same stress (0 to 300 kPa) at the variable voltage (200 to 800 V), the strain increased with the number of layers. As the number of superimposed layers increased, the total output force of the actuator increased, so the load carrying capacity increased and, thus, the negative effect of load on the actuating strain decreased accordingly. The performance of our printed actuators is better than that of the actuators made by casting reported in the literature [37].

## 4. Conclusions

In summary, we used direct writing to print a series of flexible PVC gel artificial muscles with good actuating performance in the integrated way. Inks with tailored rheological properties met the corresponding functional requirements. PVC gel ink, F-127 gel ink, CNT/SMP composite ink, and Ecoflex ink were used for printing core layers, sacrificial layers, electrode layers, and insulating layers, respectively. The inks were prepared, characterized, and evaluated for curing. Using these inks, multi-material composite artificial muscle structures with complex geometries were fabricated. Performance testing of these structures demonstrated the feasibility of the proposed manufacturing method. Our flexible PVC gel artificial muscles with good actuating performance prepared by integrated printing have potential application in soft robotics, medical rehabilitation, and wearable electronic devices. The proposed method of integrated printing artificial muscles introduces a new route to solve the inherent problems of the traditional fabricating methods of PVC gel, which lays a foundation for the broad application of this new artificial muscle material in various fields.

## Figures and Tables

**Figure 1 polymers-13-02734-f001:**
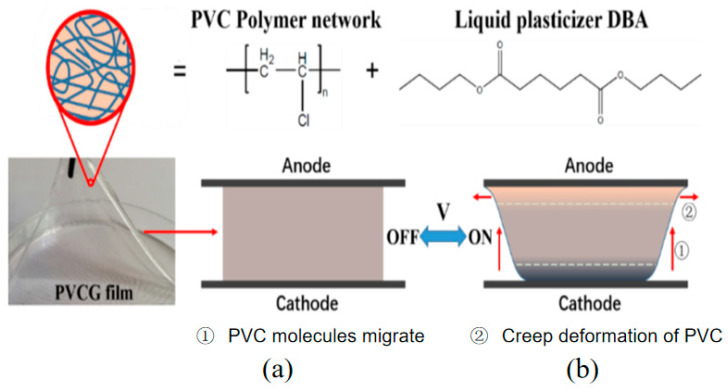
Deformation principle of PVC-gel actuator. (**a**) Discharge (**b**) Charge.

**Figure 2 polymers-13-02734-f002:**
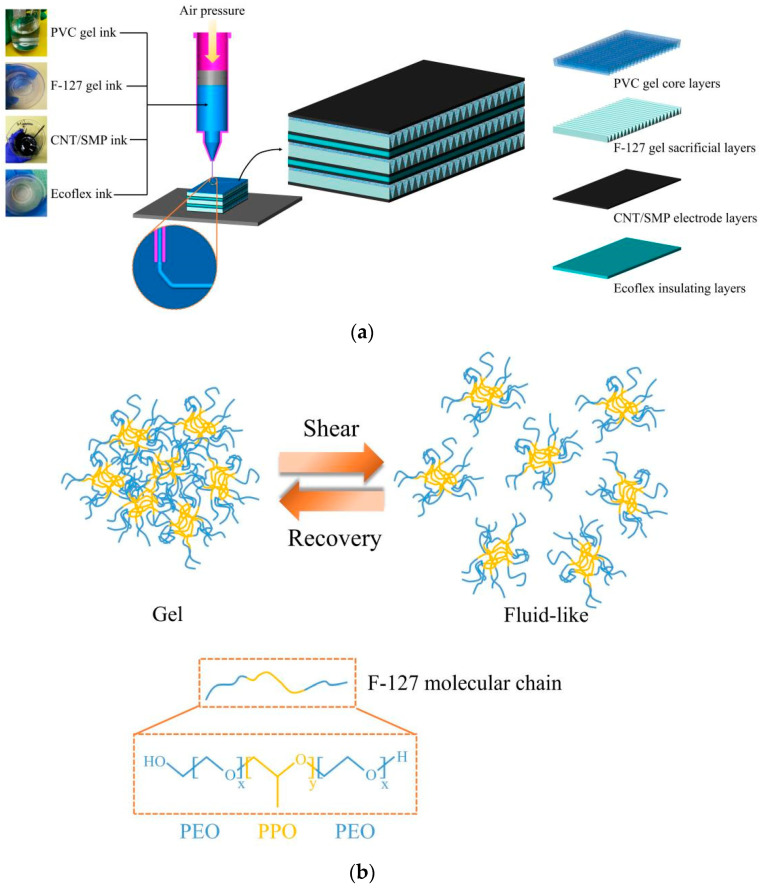
(**a**) Schematic diagram of the printing process of the artificial muscle structure. (**b**) Schematic diagram of the thixotropic property of the F-127 gel ink.

**Figure 3 polymers-13-02734-f003:**
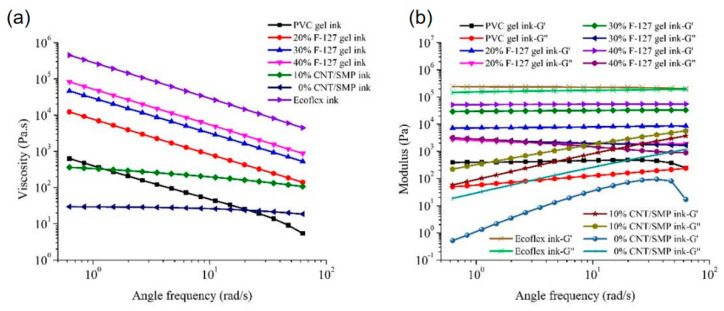
Preparation and performance characterization of the inks. (**a**) Viscosity and (**b**) moduli (storage modulus G′ and loss modulus G″) of the PVC gel ink, F-127 gel ink with different concentrations, pure SMP ink, CNT/SMP ink, and Ecoflex ink as a function of the angular frequency.

**Figure 4 polymers-13-02734-f004:**
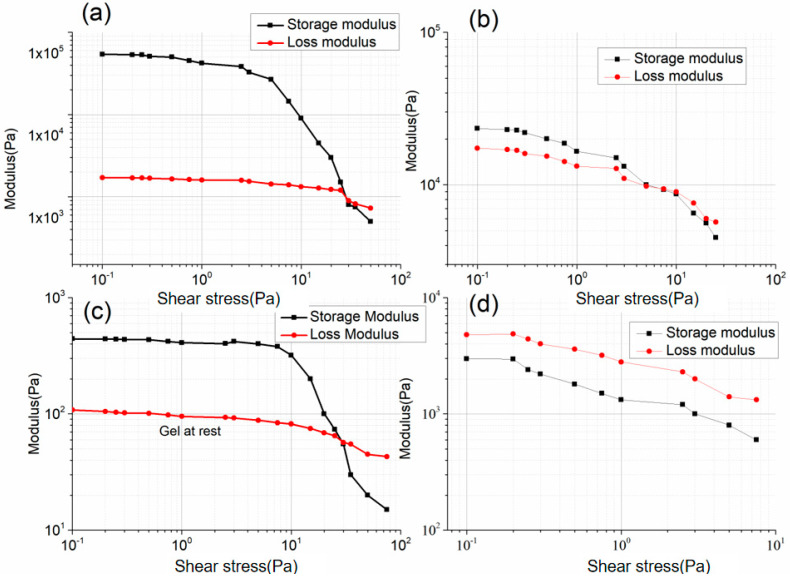
Plot of the storage modulus and loss modulus as a function of shear stress. (**a**) PVC-gel ink, (**b**) F-127 gel ink, (**c**) Ecoflex ink, (**d**) CNT/SMP composite ink.

**Figure 5 polymers-13-02734-f005:**
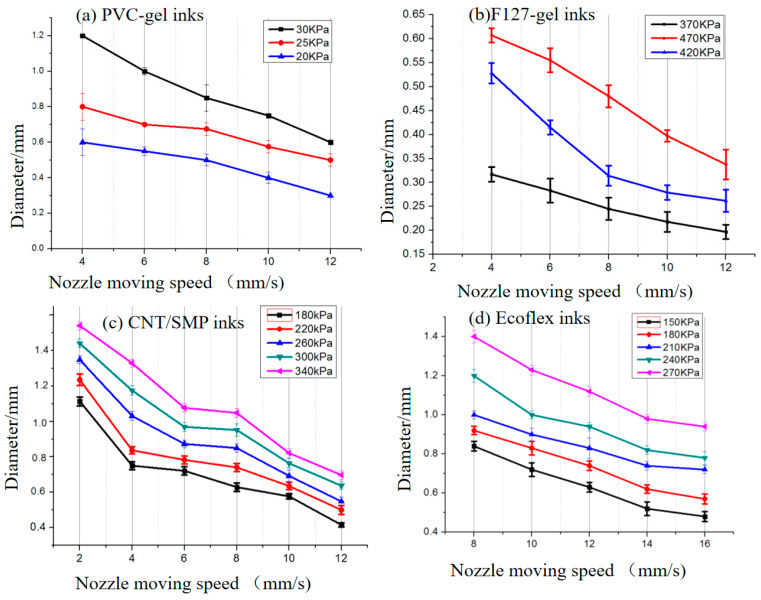
Influence of the extrusion pressure and nozzle moving speed on the printed diameter (**a**) PVC gel ink, (**b**) F-127 gel ink, (**c**) CNT/SMP ink, (**d**) Ecoflex ink.

**Figure 6 polymers-13-02734-f006:**
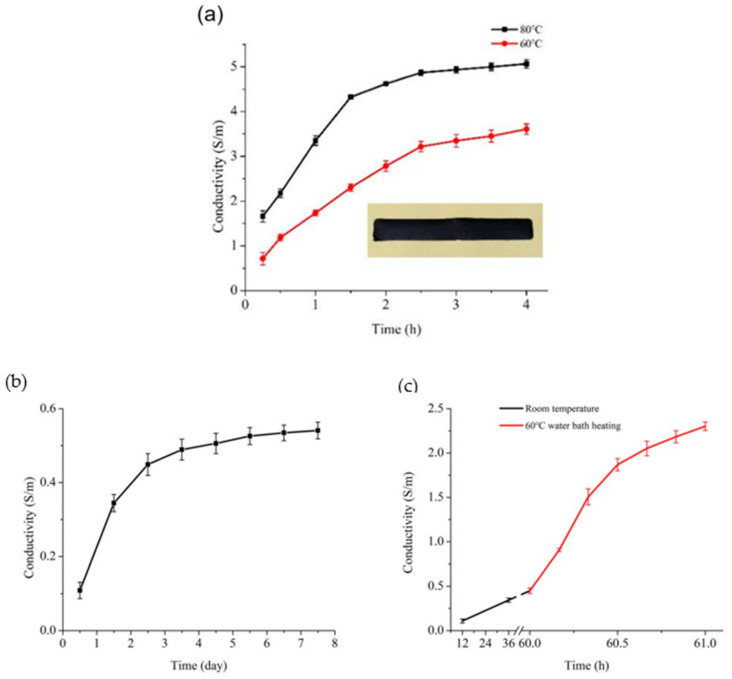
The conductivity of the CNT/SMP composite inks as functions of time (**a**) at 80 °C, 60 °C, (**b**) room temperature (25 °C), and (**c**) stepwise curing conditions.

**Figure 7 polymers-13-02734-f007:**
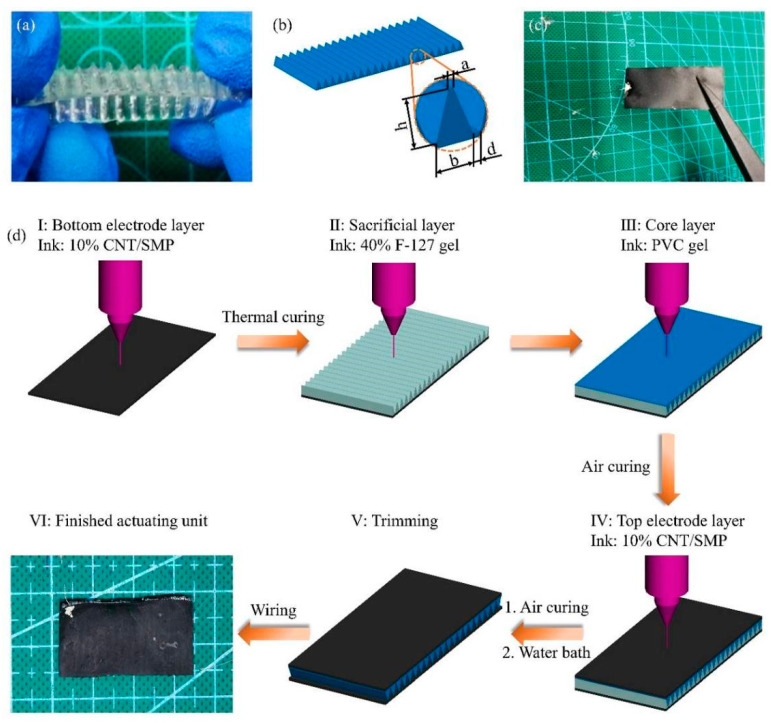
Printing process and printed structures. (**a**) Finished corrugated PVC gel core layer. (**b**) Designed shape of the corrugation in core layers. (**c**) Finished CNT/SMP composite electrode layer. (**d**) Printing process of a single layer PVC gel actuator. (**e**) Printing process of a multilayer PVC gel actuator.

**Figure 8 polymers-13-02734-f008:**
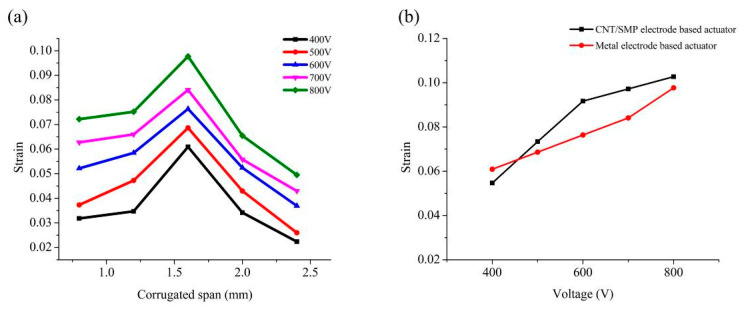
Performance of the printed structures. (**a**) The strain–span curves of metal electrode-based PVC gel actuators at different voltages. (**b**) The strain–voltage curves of the CNT/SMP composite electrode–based actuator and a metal electrode-based actuator with the same kind of core layer. (**c**) The strain–stress curves of the PVC gel actuators with different numbers of layers under voltage from 200 V to 800 V.

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
