# Peer review of "Direct Writing Corrugated PVC Gel Artificial Muscle via Multi-Material Printing Processes"

_polymers, 2021, doi:10.3390/polym13162734_

Round 1

Reviewer 1 Report

Luo and co-workers describe the preparation of a multi-material actuator composed of several layers including a (i) PVC gel core, a (ii) F-127 gel sacrificial layer, a (iii) CNT/SMP electrode and a (iv) ecoflex insulating layer. The actuator layers were produced by direct writing in a series of printing processes. The performance of the actuator was demonstrated and a series of strain-span, strain-voltage and strain-load curves are presented. The materials are not new but the disclosed method, including the use of F-127, is interesting. One concern is that the topic of the article is only a poor match to the scope of Polymers. Additionally, the authors should revise the following:

- The use of adjectives such as “impressive” (line 368) should be avoided

- The authors mention that F-127 gel ink had high printing accuracy (line 110) but the accuracy is not characterised. The authors should reconsider this aspect or modify the sentence

- Yield-stress materials are typically used in 3D printing so that sufficient accuracy in the printing process is achieved. In addition to the viscosity and the dependence on frequency of the storage modulus and loss modulus, the yield stress of the inks in the current study should be characterised  

Thus, I cannot recommend the publication of polymers-1322602 for the reasons I have mentioned above.

Author Response

We would like to thank you for giving us a chance to rewrite the paper, and also thank your suggestions which would help us to improve the quality of the paper.

We list a point-to-point response of your suggestions  in the upload word. Please check them out and give me more chance.

Reviewer 2 Report

Polymers-1322602

Title: Direct writing corrugated PVC gel artificial muscle via multi-material printing processes

The authors developed a PVC gel-based artificial muscle by direct writing a sequence of materials. Inks with proper rheological properties were developed for this purpose. The printing of PVC-gel solution appeared to be the specific contribution of this work. Nevertheless, the authors should point out the novelty of this work more clearly. There are major issues that needs to be addressed:
1. The authors reported the strain-voltage and strain-load relationships of the artificial muscle. The latter was conducted at 800 V. The authors should replace load by stress and strain-stress at other applied voltages should be reported as the loading capacity is the key characteristic of an artificial muscle.
2. Line 96: Vendor information of CNT, SMP, Ecoflex silicone, PVC, pluronic F-127, and many other chemicals used in this study should be provided. What is the exact SMP? The specifications of the printing device with a pneumatic ink extrusion system used in this study should be provided as well. The information is needed for interested readers to be able to reproduce the results.
3. The authors should compare the built artificial muscle with other similar PVC gel-based artificial muscle in the literature; related discussion is completely missing.

Minor issues include:
1. Line 19: it may be better to list the structural parameters.
2. Lines 13-14 and 21: The authors pointed out the inherent shortcoming of traditional manufacturing methods of PVC gel actuators but gave no clue of the problems. The shortcomings of traditional manufacturing methods should be briefly introduced fitting into the abstract.
3. Lines 76-78: multiple references should be cited for multiple reports; [11] is not a review article.
4. Line 82, The word conversely is confusing. Consider replacing with other appropriate adverbs. 
5. The English needs to be polished significantly; hard to believe that it has been edited by English editing service.

Author Response

We would like to thank you for giving us a chance to revise the paper, and also thank your suggestions which would help us to improve the quality of the paper. Here we present a new version of our manuscript which has been carefully modified according to your’ suggestions as highlighted in yellow.

We list a point-to-point response of the comment in the word. Please check them out.

We list a point-to-point response of the two reviewers in the next page. Please check them out.

Round 2

Reviewer 1 Report

Thanks for addressing some of the initial concerns. The quality of the manuscript has improved but there are still some aspects which have not been addressed (please see attachment - comments in blue font). Thus, I cannot recommend the publication of polymers-1322602 (revised version).

Author Response

Dear reviewer, I’m sorry for that at first don't understand what you mean.Thanks for your suggestions which would help us to improve the quality of the paper. 

Here we present a new version of our manuscript which has been carefully modified according to the your  suggestions as highlighted in yellow.

Best regards

Reviewer 2 Report

Thank you for addressing the concerns. The manuscript is improved significantly. There are a few minor issues as follows.

The authors might consider providing information on how the multi-nozzle direct writing device works. What is the working variable controlled by the PC by what mechanism? In addition, the following might be clarified. Do the multiple nozzles work simultaneously? What is the purpose of the heating platform?

Author Response

Dear reviewer, Your suggestions are sincere.Thanks for your suggestions which would help us to improve the quality of the paper.

Here we present a new version of our manuscript which has been carefully modified according to the your suggestions as highlighted in yellow. 

Best regards
